# Metrics to quantify the importance of mixing state for CCN activity

Joseph Ching[1*], Jerome Fast[1], Matthew West[2], and Nicole Riemer[3]

[1]Atmospheric Sciences and Global Change Division, Pacific Northwest National Laboratory, Richland, WA, 99354
[*]now at Meteorological Research Institute, Japan Meteorological Agency, Tsukuba, Japan
[2]Department of Mechanical Science and Engineering, University of Illinois at Urbana-Champaign, 1206 W. Green St., Urbana, IL 61801, USA
[3]Department of Atmospheric Sciences, University of Illinois at Urbana-Champaign, 105 S Gregory St., Urbana, IL 61801, USA

*Correspondence to:* J. Ching (pingpui.ching@pnnl.gov)

**Abstract.** It is commonly assumed that models are more prone to errors in predicted CCN concentrations when the aerosol populations are externally mixed. In this work we investigate this assumption by using the mixing state index ($\chi$) proposed by Riemer and West (ACP, 13, 11423-11439, 2013) to quantify the degree of external and internal mixing of aerosol populations. We combine this metric with particle-resolved model simulations to quantify error in CCN predictions when mixing state information is neglected, exploring a range of scenarios that cover different conditions of aerosol aging. We show that mixing state information does indeed become unimportant for more internally-mixed populations, more precisely for populations with $\chi$ larger than 60%. For more externally-mixed populations ($\chi$ below 20%) the relationship of $\chi$ and the error in CCN predictions is not unique, and ranges from lower than $-40\%$ to about 150%, depending on the underlying aerosol population and the environmental supersaturation. We explain the reasons for this behavior with detailed process analyses.

## 1 Introduction

The mixing state of an aerosol population depends on the distribution of chemical compounds across the population (Riemer and West, 2013). Field observations reveal that ambient aerosol mixing states can be complex (e.g. Healy et al., 2013; Moffet and Prather, 2009). Even freshly emitted particles can contain multiple chemical species depending on the source characteristics (Ault et al., 2010; Toner et al., 2006), and the initial particle composition is further modified in the atmosphere as a result of aging processes such as coagulation, condensation of secondary aerosol species, and heterogeneous reactions (Weingartner et al., 1997). This profoundly impacts the aerosol optical properties (Jacobson, 2001), their CCN activity (Wang et al., 2010) and the particle lifetime (Koch et al., 2009).

In this study we focus on CCN activity, and indeed, many CCN closure studies show that the quality of the closure depends crucially on the assumptions about aerosol mixing state (McFiggans et al., 2006; Wang et al., 2010; Bhattu and Tripathi, 2015; Ervens et al., 2010). Based on these observational findings, it is commonly assumed that the "internal mixture assumption" works well for regions that are not directly influenced by fresh emission sources. By "internal mixture assumption" we mean the assumption that the composition of individual aerosol particles equals the composition of the bulk, at least within a certain size range. In contrast, closure studies show that, in areas close to emission sources, a certain degree of external mixing needs

to be assumed to obtain good closure. This is confirmed by modeling studies showing that, without introducing fresh emissions, aging processes transform the mixing state of aerosol populations so that the CCN properties can be deduced from the bulk aerosol composition, and mixing state details become negligible (Zaveri et al., 2010; Ching et al., 2012; Fierce et al., 2013). However, the timescale for this transformation depends on the local conditions, such as total number concentration of existing CCN and the amount of condensable aerosol material (Fierce et al., 2015). Introducing fresh emissions additionally complicates the picture.

The importance of aerosol mixing state to CCN concentration is also demonstrated by studies that experimentally determine the hygroscopicity of organic species. For example, Mei et al. (2013) showed for the CalNex field campaign that the uncertainty in the derived organic hygroscopicity depended on the uncertainties in the derived CCN hygroscopicity and the volume fraction of chemical species contained in the CCN, which are related to the aerosol mixing state.

Our aim for this study is to quantitatively explain how mixing state and error in CCN concentrations are related. Previous work has approached this question from a Lagrangian point of view, considering the aging history of the particle population (Zaveri et al., 2010; Ching et al., 2012, 2016a) as a plume of aerosol particles is evolving. While this approach yields valuable process-level understanding, it is difficult to apply to field observations, because following a particle population in the real atmosphere is inherently challenging. In this study, we therefore will not focus on the temporal evolution of the particle population, but instead individually sample populations from a set of simulations. The central question that we address is: For aerosol populations of a given mixing state, what magnitude of errors can we expect for estimating CCN concentrations when assuming that the population is internally mixed?

Our study combines a recently developed metric for aerosol mixing state with particle-resolved modeling and a strategy for error quantification. We use the mixing state index ($\chi$) proposed by Riemer and West (2013) to rigorously quantify the degree of external/internal mixing of aerosol populations. This mixing state index is a scalar quantity, and varies between 0% (for completely external mixtures) and 100% (for completely internal mixtures) for any given aerosol population. The metric has been applied to field observations in Paris during the MEGAPOLI campaign (Healy et al., 2014), in Northern California during CARES (O'Brien et al., 2015), and in central London (Giorio et al., 2015). It has proven useful to gain insight into the processes that govern diurnal changes in mixing state and mixing state changes related to air mass origin. For example, Healy et al. (2014) were able to determine a characteristic diurnal cycle of $\chi$ for Paris with low values during the day when daytime primary emissions dominated, compared to the night when formation of ammonium nitrate moved the population towards a more internal mixture, indicated by increasing $\chi$ values.

Sections 2 and 3 give a brief background on the definition of the mixing state index and on the particle-resolved model PartMC-MOSAIC. Section 4 describes the model simulations that are the basis of this work and our framework for error quantification, followed by the results in Section 5 and 6. Section 7 summarizes our findings.

## 2 Mixing state metrics

Riemer and West (2013) put forward a framework to quantify aerosol mixing state, which was inspired by diversity metrics used in other disciplines such as ecology (Whittaker, 1972), economics (Drucker, 2013), neuroscience (Strong et al., 1998), and genetics (Falush et al., 2007). The salient points are summarized as follows. Given a population of $N$ aerosol particles, each consisting of some amounts of $A$ distinct aerosol species, this concept is based on the knowledge of mass of species $a$ in particle $i$, denoted $\mu_i^a$, for $i = 1, \ldots, N$, and $a = 1, \ldots, A$. From this quantity, all other mass-related quantities can be defined, as detailed in Riemer and West (2013) and here listed in Table 1, and the diversity metrics can be constructed as shown in Table 2.

The particle diversity $D_i$ represents the number of "effective species" of particle $i$. For a particle $i$ that consists of $A$ species, the highest possible value for $D_i$ is $A$, and this occurs when all $A$ species are present in equal mass fractions. Knowing the $D_i$ values for all particles, the population-level quantities $D_\alpha$ and $D_\gamma$ can be calculated, with $D_\alpha$ being the average effective number of species in each particle, and $D_\gamma$ being the effective number of species in the bulk. Finally, the mixing state index $\chi$ is defined as

$$\chi = \frac{D_\alpha - 1}{D_\gamma - 1}. \tag{1}$$

The mixing state index $\chi$ varies from 0% (a fully externally mixed population) to 100% (a fully internally mixed population. To fully quantify mixing state, two of the three metrics $(D_\alpha, D_\gamma, \chi)$ are needed, and the third can be derived. As shown in Riemer and West (2013) and Healy et al. (2014), it is instructive to map the mixing state metrics of aerosol populations into a mixing state diagram $(D_\alpha, D_\gamma)$, as shown in Figure 1. Particle populations with single-species particles, i.e., "externally mixed" populations, have $D_\alpha = 1$ and $D_\gamma$ between 1 and $A$, and are therefore mapped onto the vertical axis ($\chi = 0\%$). Populations consisting of particles with identical mass fractions map onto the diagonal $\chi = 100\%$. Since $\chi$ has the intuitive interpretation of the "degree of internal mixing", we use it here as a metric for error quantification as shown in Section 4.

Note that the definition of "species" for calculating the mass fractions depends on the application. It can refer to chemical species, as in Riemer and West (2013), Healy et al. (2014), O'Brien et al. (2015), and Giorio et al. (2015), or it can refer to species groups, as in Dickau et al. (2016) who quantified mixing state with respect to volatile and non-volatile components. Since we are concerned with CCN properties in this paper, we will group the chemical model species according to hygroscopicity, defining two surrogate species. Black carbon (BC) and primary organic aerosol (POA) are combined into one surrogate species, since their hygroscopicity is very low. All other model species (inorganic and secondary organic aerosol species) are combined into a second surrogate species and $\chi$ is calculated from these two surrogate species.

## 3 Particle-resolved modeling with PartMC-MOSAIC

A detailed description of the numerical methods used in PartMC-MOSAIC is given in Riemer et al. (2009). In brief, PartMC (Particle-resolved Monte Carlo) is a 0-D, or box, model which explicitly resolves the composition of many individual particles within a well-mixed computational volume representing a much larger air parcel. During the evolution of the air parcel moving

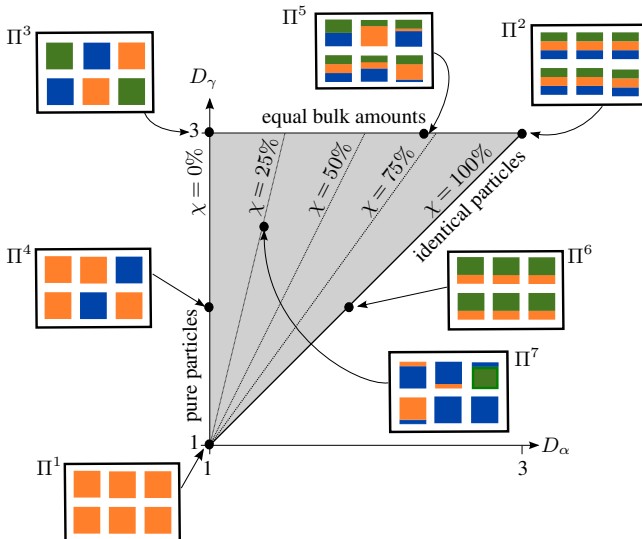

**Figure 1.** Mixing state diagram to illustrate the relationship between average particle diversity $D_\alpha$, bulk population diversity $D_\gamma$, and mixing state index $\chi$ for seven example populations ($\Pi_1$ to $\Pi_7$). Each population consists of six particles, and the colors represent different chemical species ($A = 3$). This figure is taken from Riemer and West (2013).

| Quantity | Meaning |
|---|---|
| $\mu_i^a$ | mass of species $a$ in particle $i$ |
| $\mu_i = \sum_{a=1}^{A} \mu_i^a$ | total mass of particle $i$ |
| $\mu^a = \sum_{i=1}^{N} \mu_i^a$ | total mass of species $a$ in population |
| $\mu = \sum_{i=1}^{N} \mu_i$ | total mass of population |
| $p_i^a = \dfrac{\mu_i^a}{\mu_i}$ | mass fraction of species $a$ in particle $i$ |
| $p_i = \dfrac{\mu_i}{\mu}$ | mass fraction of particle $i$ in population |
| $p^a = \dfrac{\mu^a}{\mu}$ | mass fraction of species $a$ in population |

**Table 1.** Aerosol mass and mass fraction definitions and notation. The number of particles in the population is $N$, and the number of species is $A$. This table is taken from Riemer and West (2013)

.

| Quantity | Name | Units | Range | Meaning |
|---|---|---|---|---|
| $H_i = \sum_{a=1}^{A} -p_i^a \ln p_i^a$ | mixing entropy of particle $i$ | — | 0 to $\ln A$ | Shannon entropy of species distribution within particle $i$ |
| $H_\alpha = \sum_{i=1}^{N} p_i H_i$ | average particle mixing entropy | — | 0 to $\ln A$ | average Shannon entropy per particle |
| $H_\gamma = \sum_{a=1}^{A} -p^a \ln p^a$ | population bulk mixing entropy | — | 0 to $\ln A$ | Shannon entropy of species distribution within population |
| $D_i = e^{H_i} = \prod_{a=1}^{A} (p_i^a)^{-p_i^a}$ | particle diversity of particle $i$ | effective species | 1 to $A$ | effective number of species in particle $i$ |
| $D_\alpha = e^{H_\alpha} = \prod_{i=1}^{N} (D_i)^{p_i}$ | average particle (alpha) species diversity | effective species | 1 to $A$ | average effective number of species in each particle |
| $D_\gamma = e^{H_\gamma} = \prod_{a=1}^{A} (p^a)^{-p^a}$ | bulk population (gamma) species diversity | effective species | 1 to $A$ | effective number of species in the bulk |
| $D_\beta = \dfrac{D_\gamma}{D_\alpha}$ | inter-particle (beta) diversity | — | 1 to $A$ | amount of population species diversity due to inter-particle diversity |
| $\chi = \dfrac{D_\alpha - 1}{D_\gamma - 1}$ | mixing state index | — | 0% to 100% | degree to which population is externally mixed ($\chi = 0\%$) versus internally mixed ($\chi = 100\%$) |

**Table 2.** Definitions of aerosol mixing entropies, particle diversities, and mixing state index. In these definitions we take $0\ln 0 = 0$ and $0^0 = 1$. This table is taken from Riemer and West (2013)

.

along a specific trajectory, the mass of each constituent species within each particle is tracked. Emission, dilution, nucleation and Brownian coagulation are simulated with a stochastic Monte Carlo approach. The relative positions of particles within the computational volume are not tracked. To improve efficiency of the method we use weighted particles in the sense of DeVille et al. (2011) and efficient stochastic sampling methods (Michelotti et al., 2013).

PartMC is coupled with the aerosol chemistry model MOSAIC (Model for Simulating Aerosol Interactions and Chemistry) (Zaveri et al., 2008) which includes the gas phase photochemical mechanism CBM-Z (Zaveri and Peters, 1999), the Multicomponent Taylor Expansion Method (MTEM) for estimating activity coefficients of electrolytes and ions in aqueous solutions (Zaveri et al., 2005a), the multi-component equilibrium solver for aerosols (MESA) for intraparticle solid-liquid partitioning (Zaveri et al., 2005b) and the adaptive step time-split Euler method (ASTEM) for dynamic gas-particle partitioning over size- and composition-resolved aerosol (Zaveri et al., 2008), as well as a treatment for SOA (secondary organic aerosol) based on the SORGAM scheme (Schell et al., 2001). The CBM-Z gas phase mechanism treats a total of 77 gas species. MOSAIC treats key aerosol species including sulfate ($SO_4$), nitrate ($NO_3$), ammonium ($NH_4$), chloride (Cl), carbonate ($CO_3$), methanesulfonic acid (MSA), sodium (Na), calcium (Ca), other inorganic mass (OIN), BC, POA, and SOA. The model species OIN represents species such as $SiO_2$, metal oxides, and other unmeasured or unknown inorganic species present in aerosols. SOA includes
reaction products of aromatic precursors, higher alkenes, $\alpha$-pinene and limonene.

PartMC-MOSAIC has been used in the past for process studies of mixing state impacts on aerosol properties in various environments. For example, Ching et al. (2012) quantified the impact of aerosol mixing state on cloud droplet formation, and Fierce et al. (2013) investigated the sensitivity of CCN activity to mixing state characteristics at emission. The model was also used to explain the observed diurnal variations of aerosol hygroscopicity in the North China Plain (Liu et al., 2011) and to
characterize the evolution of aerosol mixing state in a ship plume (Tian et al., 2014).

## 4   Framework for error quantification

The basis for the error quantification framework is the library of eight urban plume scenarios described in Ching et al. (2016a), Sections 3.2.1 and 4.1. These scenarios were designed to simulate the aging process of black carbon-containing particles from combustion sources. With "aging" we refer here to the transition from CCN-inactive to CCN-active at a given supersaturation
threshold due to coagulation with other aerosol particles or due to condensation of secondary aerosol material (Riemer et al., 2010). We consider the formation of ammonium nitrate and ammonium sulfate as well as secondary organic aerosol from biogenic or anthropogenic gaseous precursors as represented in the SORGAM module.

The eight scenarios are derived from the base case, which is the scenario presented in Ching et al. (2012). To construct the scenario library, the BC emission rate was set to 100% (E100), 25% (E25), and 2.5% (E2.5) of the base-case, and the number
concentration of background aerosol particles was set to 100% (B100), and 10% (B10) of the base-case. Combining the emission cases and the background aerosol concentration cases results in six scenarios, and the gas phase emissions for these six scenarios were the same as in Ching et al. (2012). For the base case (E100-B100) we performed two additional simulations by reducing the emission rate of all gaseous components by 50% (G50) and 25% (G25), respectively, of the case presented

| Initial/Background | $N_a$ / cm$^{-3}$ | $D_g$ / $\mu$m | $\sigma_g$ | Composition by mass |
|---|---|---|---|---|
| Aitken mode | 1800 | 0.02 | 1.45 | 49.64% (NH$_4$)$_2$SO$_4$ + 49.64% SOA + 0.72% BC |
| Accumulation mode | 1500 | 0.116 | 1.65 | 49.64% (NH$_4$)$_2$SO$_4$ + 49.64% SOA + 0.72% BC |

| Emission | $E_a$ / m$^{-2}$ s$^{-1}$ | $D_g$ / $\mu$m | $\sigma_g$ | Composition by mass |
|---|---|---|---|---|
| Meat cooking | $9 \times 10^6$ | 0.086 | 1.91 | 100% POA |
| Diesel Vehicles | $1.6 \times 10^8$ | 0.05 | 1.74 | 30% POA + 70% BC |
| Gasoline Vehicles | $5 \times 10^7$ | 0.05 | 1.74 | 80% POA + 20% BC |

**Table 3.** Number concentration, $N_a$, of the initial/background aerosol population, and area source strength, $E_a$, of the three types of emission. All aerosol size distributions are assumed to be lognormal and defined by the geometric mean diameter, $D_g$, and the geometric standard deviation, $\sigma_g$.

in Ching et al. (2012). The BC mass concentrations of all the plume scenarios ranged from 0.05 $\mu$g m$^{-3}$ to 3.6 $\mu$g m$^{-3}$. The formation of secondary aerosol material was similar in all G100 cases, since the emissions of gaseous precursors were the same. On the first simulation day, the total mass concentration was dominated by ammonium nitrate formation, which evaporated after ammonia emissions ceased (Ching et al., 2012). Secondary production of sulfate and organic mass occurred

on the first and second days of the simulation. The total mass concentration varied by a factor of about two between plume scenarios G100 and G50 due to differences in secondary aerosol mass formation.

As a result of these changes in input parameters, the aging of BC particles proceeded at different rates, and different mixing states arose over the course of the simulation, which are further discussed below. The simulation time for each scenario was 48 hours, and during the first 12 hours gas and aerosol emissions entered the air parcel. Background aerosol particles were

introduced over the entire simulation time owing to dilution with background air. Table 3 specifies the details of the initial and background aerosol distribution as well as the aerosol emissions for the base case run.

Here we are not focused on the process analysis of the temporal evolution of BC-containing particles, but rather on the aerosol state; we use the hourly aerosol state from the scenarios as a basis for our analysis. We will refer to the set of these populations as set $\mathbb{P}$, which comprises $N_{pop} = 48 \times 8 = 384$ elements. The populations in $\mathbb{P}$ cover a wide range of mixing

states. Figure 2 illustrates how these aerosol populations map to the mixing state diagram. Each symbol represents an aerosol population from $\mathbb{P}$, colored by the scenario the population is sampled from.

As explained in Section 2, the maximal possible range for both $D_\alpha$ and $D_\gamma$ is from 1 to 2, however this range is not entirely accessed with the populations investigated here. The effective number of species in the population ($D_\gamma$) is always larger than 1, i.e., there are no populations that consist solely of hygroscopic or hydrophobic species. It reaches values of almost 2 for some

populations, meaning that for these populations the bulk mass fractions of hydrophobic and hygroscopic material are about the same. On a per-particle level, the average number of effective species ($D_\alpha$) is close to 1 for some populations, indicating that populations exist for which the particles consist of purely hygroscopic or hydrophilic material. The maximum value of $D_\alpha$ is about 1.7. Figure 2 also shows lines of constant mixing state index $\chi$. The populations cover values of $\chi$ from 7.5%

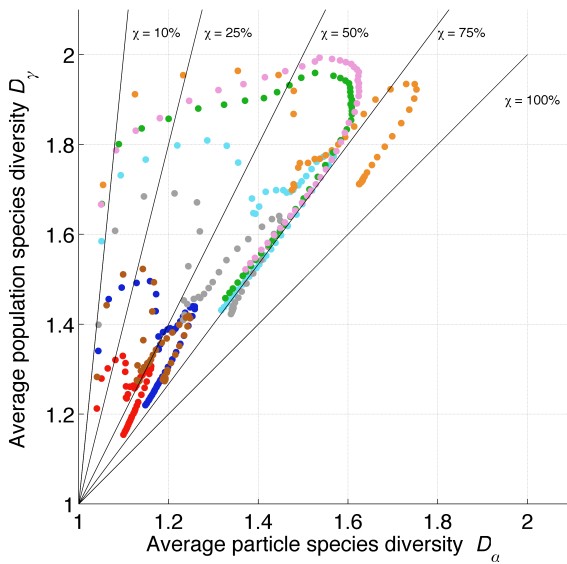

**Figure 2.** Mixing state diagram for the 384 populations in $\mathbb{P}$. The colors indicate the Ching et al. (2016a) scenario from which the populations were sampled: B100-E100-G100 (light blue), B10-E100-G100 (orange), B100-E25-G100 (blue), B10-E25-G100 (grey), B100-E2.5-G100 (red), B10-E2.5-G100 (brown), B100-E100-G50 (green) and B100-E100-G25 (pink). The abbreviations of the scenarios follow Ching et al. (2016a), "B" indicates the level of background particle concentration (100% and 10% of the base case scenario), "E" indicates the level of the black carbon emission rate (100%, 25%, and 2.5% of the base case scenario) and "G" indicates the level of gas emission rate (100%, 50%, and 25% of the base case scenario).

to 88%. The lack of populations with $\chi$ between 88% and 100% does not impact the generality of the conclusions. As we will see later, the error in CCN concentration when neglecting mixing state information vanishes for $\chi > 60\%$. The pattern that each scenario forms in this mixing state diagram can be explained by the temporal evolution of $D_\gamma$ and $D_\alpha$, which is determined by coagulation, condensation of secondary aerosol material, dilution, and emission (Riemer and West, 2013). All

5    scenarios start with low average particle species diversity, $D_\alpha$, because at the beginning of the simulation the particles consist mainly of one of the surrogate species, either hydrophobic BC or POA, or hygroscopic background species. As the simulation proceeds, secondary aerosol species condense and coagulation occurs, moving $D_\alpha$ towards larger values. This process does not necessarily proceed monotonically, i.e., $D_\alpha$ can also decrease with time. The reason could be either that secondary aerosol material is evaporating or that so much secondary aerosol material is condensing that it dominates in terms of composition. A

10   similar argument can be made for the temporal evolution of $D_\gamma$, but applied to the composition of the bulk population.

   For each of the 384 aerosol populations we determine CCN concentrations as follows: Since we track the composition evolution of each individual particle throughout the simulation, we can calculate the critical supersaturation $s_c$ for each particle as

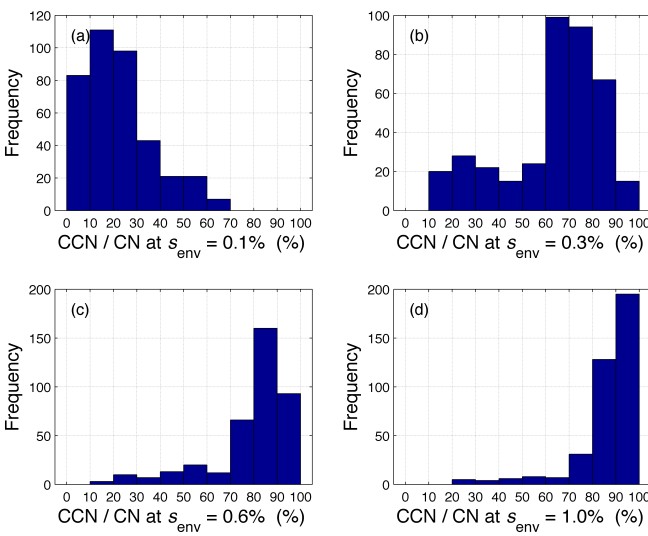

**Figure 3.** Frequency distribution of CCN/CN fractions for all 384 aerosol populations in $\mathbb{P}$ at four selected environmental supersaturations $s_{\mathrm{env}}$. The 384 populations were derived from eight scenarios, each with 48 hourly snapshots.

described in Riemer et al. (2010), using the concept of the dimensionless hygroscopicity parameter $\kappa$ (Petters and Kreidenweis, 2007). The overall $\kappa$ for a particle is the volume-weighted average of the $\kappa$ values of the constituent species. Based on Petters and Kreidenweis (2007) we assume $\kappa = 0.65$ for all salts formed from the $\mathrm{NH_4^+ - SO_4^{2-} - NO_3^-}$ system. For all MOSAIC model species that represent SOA we assume $\kappa = 0.1$, and for POA and BC we assume $\kappa = 0.001$ and $\kappa = 0$, respectively.

We use $s_{\mathrm{c}}$ to define a cumulative number distribution $N_{\mathrm{CCN}}(s_{\mathrm{c}})$, which represents the number of particles per volume with critical supersaturation less than $s_{\mathrm{c}}$. The CCN concentration at a chosen environmental supersaturation threshold $s_{\mathrm{env}}$ for a population with mixing state index of $\chi_0$ is then denoted as $N_{\mathrm{CCN}}(s_{\mathrm{env}}, \chi_0)$.

    Figure 3 provides an overview of the CCN activity for the populations. It shows the frequency distribution of the CCN/CN fraction, evaluated at four different supersaturation thresholds (0.1%, 0.3%, 0.6%, and 1%). These four supersaturation thresh-

10 olds bound the CCN activity of our populations well. As expected, for the lowest supersaturation threshold of 0.1%, most populations only have small CCN/CN fractions with the median at $\mathrm{CCN/CN} = 20\%$. For the highest supersaturation threshold of 1% the median is at $\mathrm{CCN/CN} = 90\%$.

    We determine the error that is introduced by neglecting mixing state information as shown schematically in Figure 4. The starting point is an aerosol population from the particle-resolved simulations as shown in Figure 4(a). This particular population

had a mixing state index of $\chi_0 = 13\%$. Figure 4 shows the distribution density function $\partial^2 N(D, \kappa)/(\partial \log_{10} D \, \partial \kappa)$ based on the two-dimensional cumulative number distribution $N(D, \kappa)$ in terms of particle diameter $D$ and hygroscopicity parameter $\kappa$ (Su et al., 2010; Fierce et al., 2013). We observe that at a particular particle size a distribution of hygroscopicity parameter

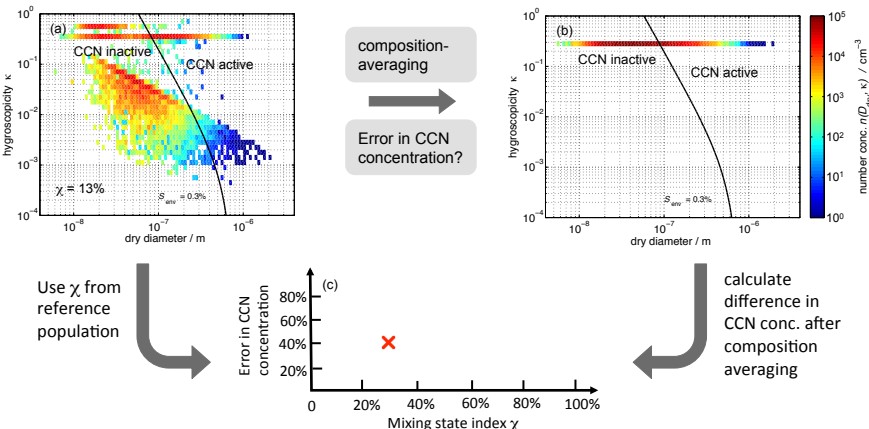

**Figure 4.** Conceptual framework for error quantification, see Section 4 for details.

values exists. The solid black line indicates the chosen supersaturation threshold (here 0.3% as an example) that separates CCN-active from CCN-inactive particles.

We then perform composition averaging on this particle population as described by Ching et al. (2016a). Composition averaging preserves the bulk aerosol mass concentrations, the number concentration and the particle diameters, but modifies

the per-particle composition, so that each particle is assigned a composition equal to the average composition of the population. Figure 4(b) shows the population after composition averaging. Because all particles were assigned the same composition (equal to the average composition), the spread in $\kappa$ vanishes, and the CCN number concentration was altered as a result.

After composition averaging we re-calculate the CCN concentration, $\bar{N}_{\mathrm{CCN}}(s_{\mathrm{env}})$, and define the relative error

$$\epsilon(s_{\mathrm{env}}, \chi_0) = \frac{\bar{N}_{\mathrm{CCN}}(s_{\mathrm{env}}) - N_{\mathrm{CCN}}(s_{\mathrm{env}}, \chi_0)}{N_{\mathrm{CCN}}(s_{\mathrm{env}}, \chi_0)}. \tag{2}$$

Finally, we graph the error $\epsilon(s_{\mathrm{env}}, \chi)$ as a function of the mixing state parameter $\chi$ for a given value of $s_{\mathrm{env}}$ of the reference population as shown in Figure 4(c).

## 5   Relationship of error in CCN concentration and mixing state index $\chi$

To aid the interpretation of the resulting error distributions as defined in Equation 2, we first show the effect of composition averaging on CCN concentrations in Figure 5 for one example population in $\mathbb{P}$. This normalized 2D histogram relates the critical

supersaturation of each particle before composition averaging to its critical supersaturation after composition averaging. Points above the 1:1 line represent particles for which the critical supersaturation increased after composition averaging, and points below the 1:1 line represent particles for which the critical supersaturation decreased after composition averaging. Figure 5 also shows a vertical and horizontal line that marks an assumed environmental supersaturation threshold of 0.3%. The particles to

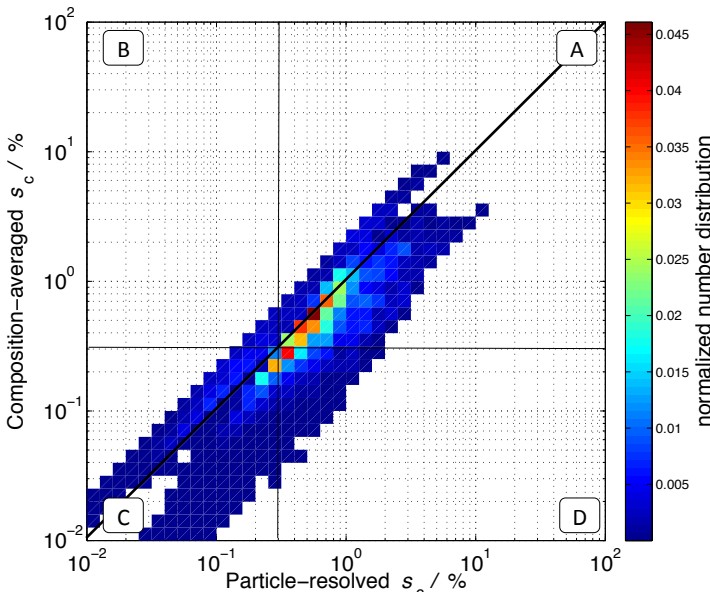

**Figure 5.** 2D histogram of particles' critical supersaturations before and after composition averaging for one aerosol population from set $\mathbb{P}$. Particles in quadrant A are not CCN active before or after composition averaging. Particles in quadrant C are CCN active before and after composition averaging. Particles in quadrant B activate in the reference case, but not after composition averaging. Particles in quadrant D activate after composition averaging, but not in the reference case.

the left of the vertical line activate in the reference case, and the particles below the horizontal line activate in the composition-averaged case.

Based on this we can define four quadrants. Particles in quadrant C activate both in the reference case and in the composition-averaged case, while in quadrant A they do not activate in either case. Particles in these two quadrants do not incur any error in

CCN concentration even though composition averaging may have changed their critical supersaturation value. Error is caused only by the *difference* in the number of particles that activate in the reference case, but do not activate after composition averaging (quadrant B), and the cases that activate after composition averaging, but did not activate in the reference case (quadrant D). If the number concentration of particles in quadrants B and D was equal, the overall error in CCN concentration would be zero even though a large number of particles might have been mis-classified.

The number concentrations in quadrants B and D are determined by the supersaturation threshold and the extent to which composition averaging simplifies mixing state, i.e., the distribution $n(D, \kappa)$ is distorted (compare Figure 4). This determines the error in the critical supersaturation distribution for the composition-averaged population compared to its particle-resolved counterpart. For populations with $\chi = 100\%$, the particles' composition is the same across the entire population, and hence all the particles have the same hygroscopicity values. Composition averaging therefore does not distort $n(D, \kappa)$ and the number

concentrations in quadrants B and D are zero. In constrast, for populations with $\chi < 100\%$, composition averaging distorts

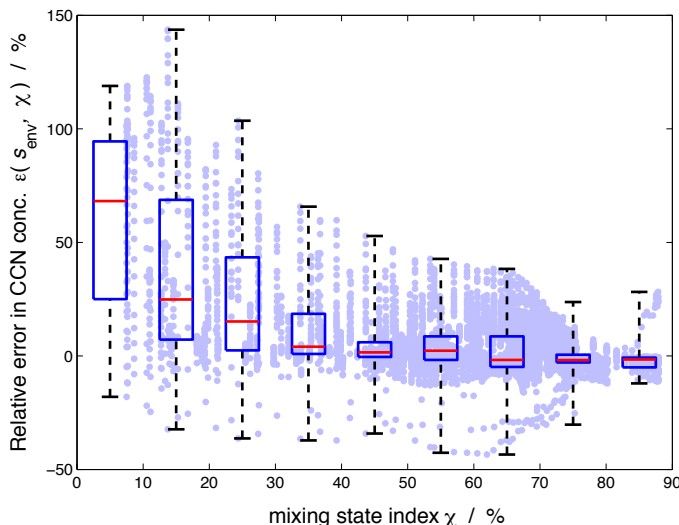

**Figure 6.** Relative error $\epsilon(s_{\mathrm{env}}, \chi)$ for each individual aerosol population in $\mathbb{P}$ (light purple dots), evaluated for 20 supersaturation values between 0.05% and 1%, see Equation 2 for definition. The boxes show the 25th and 75th percentiles of the distribution of relative errors in each $\chi$-bin. The red line is the median of that distribution, and the dashed line extends to the minimum and maximum error.

$n(D, \kappa)$ depending on the distribution of $n(D, \kappa)$. The 2D histogram in Figure 5 therefore varies from population to population, and the number concentrations in quadrants B and D depend on both the extent of distortion of the critical supersaturation distribution and the supersaturation threshold (the black horizontal and vertical lines) as shown in Figure 5.

In summary, the dependence of the relative error on supersaturation threshold and mixing state index ($\chi$) is determined by the number concentrations in quadrants B and D. Whenever the number concentrations in quadrants B and D are of similar magnitude, the error is small. As we will show in Figures 7 and 8 this can occur for all $\chi$ values. The situation of number concentrations in quadrants B and D having different magnitudes tends to occur only for small $\chi$ values.

Figure 6 shows the relative errors $\epsilon(s_{\mathrm{env}}, \chi)$ for all populations, evaluated for 20 supersaturation values ranging between 0.05% and 1% in steps of 0.05%. The boxes show the 25th and 75th percentiles of the distribution of relative errors in each $\chi$-bin ($\Delta\chi = 10\%$). The red line is the median of that distribution, and the dashed line extends to the minimum and maximum. This figure confirms that the error tends to decrease as $\chi$ increases, and that for a given $\chi$ value a range of error values exists, which narrows as $\chi$ increases as presented by the boxes in Figure 6. This range is caused by the assumed supersaturation threshold at which the CCN concentration is evaluated, but also reflects that, even for a given supersaturation threshold and a given $\chi$ value, different populations experience different amounts of the error cancellations illustrated in Figure 5.

Further, the relative error $\epsilon(s_{\mathrm{env}}, \chi)$ is positive for most populations, meaning that composition averaging causes the CCN concentration to be overestimated. However, for some populations the relative error is negative, i.e., composition averaging results in an underestimate of the CCN concentration. Cases with negative relative error mostly occur at low environmental supersaturation ($s_{\mathrm{env}} \leq 0.1\%$) and are most prevalent for $\chi$ values between 60% and 80%. For most of these cases the error

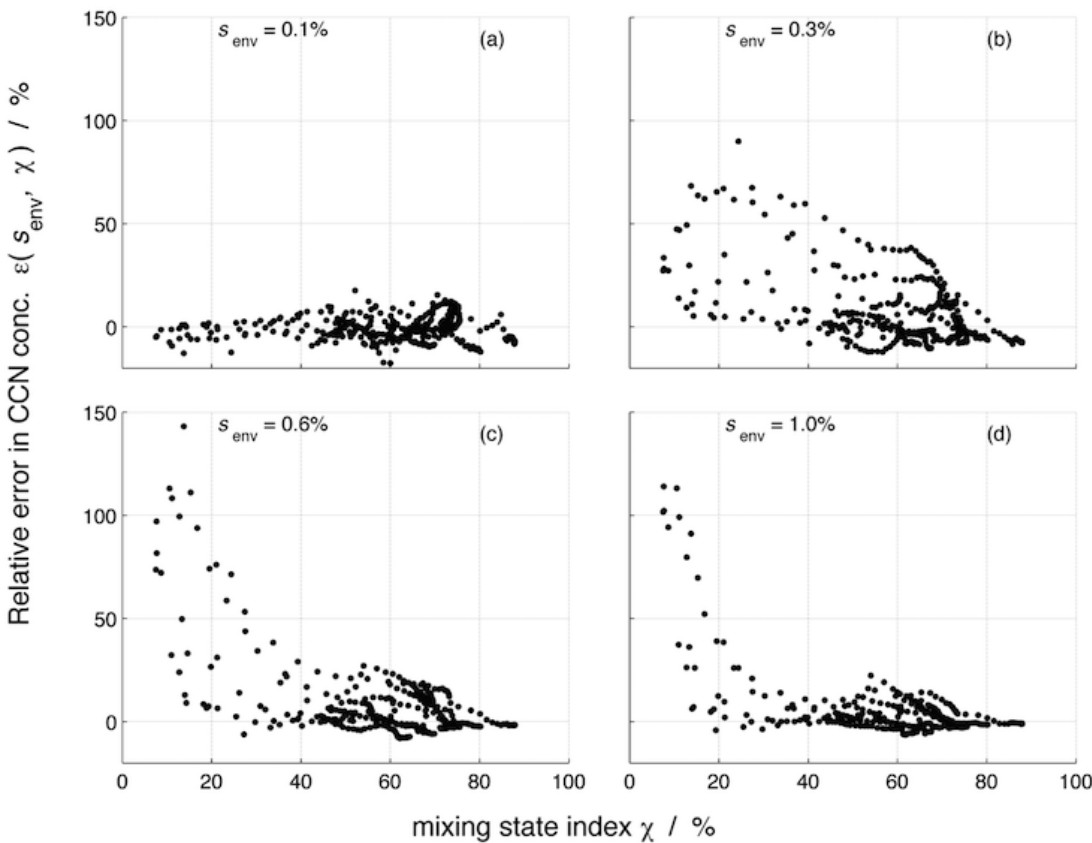

**Figure 7.** Relative error in CCN concentration $\epsilon(s_{\mathrm{env}}, \chi)$ for different environmental supersaturations, see Equation 2 for definition. The insets show the range for $\chi$ between 40% and 80% in more detail.

magnitude is less than 10% (i.e., $-10\% \le \epsilon \le 0\%$), however some points stand out with negative relative errors of up to $-43\%$. These originate from the cases with $s_{\mathrm{env}} = 0.05\%$.

To further investigate the dependence of the relative error $\epsilon(s_{\mathrm{env}}, \chi)$ on supersaturation threshold, Figure 7 shows $\epsilon(s_{\mathrm{env}}, \chi)$ for four selected environmental supersaturation thresholds. For low supersaturation thresholds ($s_{\mathrm{env}} = 0.1\%$, Figure 7(a)), the

5   error in CCN concentration is independent of the mixing state index and the magnitude is within $\pm18\%$. We can explain this fact by consulting Figure 8, which maps the number concentrations in quadrant D and quadrant B. If the concentrations are the same, the points line up on the 1:1 line and the overall error is small because of the cancellation effect explained above. Figure 8(a) shows that this is indeed generally the case for $s_{\mathrm{env}} = 0.1\%$.

At higher supersaturation thresholds (Figure 7(b), (c), and (d)), the highest errors are observed for the populations with the

10   lowest $\chi$ values, however at a given $\chi$ value a range of errors is possible. The exact outcome depends on to what extent the mis-classified particles in quadrants B and D cancel out. Errors in the intermediate regime of $\chi$ between 40% and 80% are up

to about 50%. The largest errors occur below $\chi = 20\%$ (up to about 150%). We see in Figure 8 that this behavior can again be explained with the way the quadrants B and D are populated. The points furthest away from the 1:1 line tend to have the lowest $\chi$ values. This is consistent with our expectation that accounting for mixing state is important for more "externally mixed" populations. However, some points are close to the 1:1 line even though their $\chi$ value is low, which explains why small errors can still be found for low-$\chi$ populations.

These results illustrate that the dependence of CCN concentration error on $\chi$ depends on the chosen supersaturation threshold. The reason for this can be understood from Figure 5, which shows that the change in particle number concentration per supersaturation interval is smaller for $s_c$ of 0.1% compared to $0.3 < s_c < 1.5\%$. Therefore, for the case of $s_{\mathrm{env}} = 0.1\%$, the number concentrations in B and D for this population are less sensitive to composition averaging. While this figure shows one specific population as an example, this is generally the case for our aerosol populations and explains why at low supersaturation threshold, the CCN error appears to be independent of $\chi$. As the supersaturation threshold increases, the number concentrations in B and D vary more significantly as demonstrated in Figure 5, and such variations depend on the distortion of $n(D, \kappa)$ by composition averaging. This results in a higher sensitivity of CCN error to $\chi$ at higher supersaturation thresholds. It is important to keep in mind that while this behavior applies to the aerosol populations in this study, it cannot be expected that it necessarily applies to any arbitrary aerosol population.

It should be noted here that this calculation represents an upper bound for the error, because composition averaging was performed for the entire population. If size-resolved composition information is retained, the magnitude of error decreases. The error decrease is more pronounced for higher environmental supersaturations. For example, the range of error for $s_{\mathrm{env}} = 0.1\%$ reduces from $(-18\%, +18\%)$ to $(-12\%, +16\%)$ when the particles are sorted into five size bins per decade, and composition averaging is performed within each size bin. For $s_{\mathrm{env}} = 0.3\%$ the range reduces from $(-12\%, +90\%)$ to $(-2\%, +43\%)$, and for $s_{\mathrm{env}} = 1\%$ the range reduces from $(-6\%, +114\%)$ to $(-2\%, +49\%)$. This finding is consistent with observations by Che et al. (2016). They analyzed CCN concentration measurements from the Yangtse River Delta area of China and found that at low supersaturations ($\leq 0.1\%$) using only aerosol bulk composition information was sufficient to predict CCN, whereas at higher supersaturations ($\geq 0.2\%$) using size-resolved chemical composition information improved the prediction of CCN concentrations.

To produce a summary of the above information, we calculated a normalized root mean square deviation metric as a function of mixing state index $\chi$, integrating over all supersaturations as follows:

$$\epsilon_{\mathrm{NRMSD},j} = \frac{1}{\frac{1}{n_{\mathrm{ss}} m_j} \Sigma_{m=1}^{m_j} \Sigma_{i=1}^{n_{\mathrm{ss}}} N_{\mathrm{CCN},m}(s_i)} \sqrt{\frac{\Sigma_{m=1}^{m_j} \Sigma_{i=1}^{n_{\mathrm{ss}}} \left(\bar{N}_{\mathrm{CCN},m}(s_i) - N_{\mathrm{CCN},m}(s_i)\right)^2}{n_{\mathrm{ss}} m_j}}, \tag{3}$$

where $n_{\mathrm{ss}} = 20$ is the number of supersaturation values at which we evaluated CCN concentrations, and $m_j$ is the number of aerosol populations in a given $\chi$-bin ($j = 1, \ldots, 10$). The green line in Figure 9 shows the relationship of $\epsilon_{\mathrm{NRMSD}}$ and $\chi$. Figure 9 emphasizes that composition averaging causes larger errors as $\chi$ decreases. The average error decreases from about 90% to 30% when $\chi$ increases from 0% to 30%. For $\chi$ between 40% and 90%, the average error is less than 20%. The average value $\epsilon_{\mathrm{NRMSD}}$ across the whole range of $\chi$ is 17.7%. This is comparable to the error of 14.6% found in the size-resolved simulation by Ching et al. (2016b).

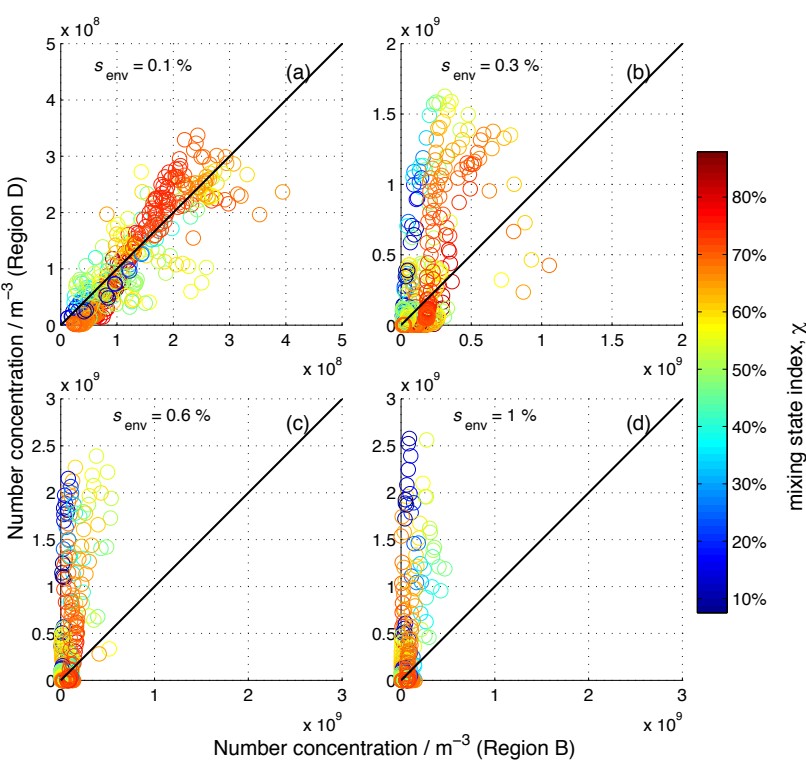

**Figure 8.** Relationship of number concentration in quadrants B and D (compare Figure 5) for different environmental supersaturations. The data points are colored corresponding to the mixing state index $\chi$.

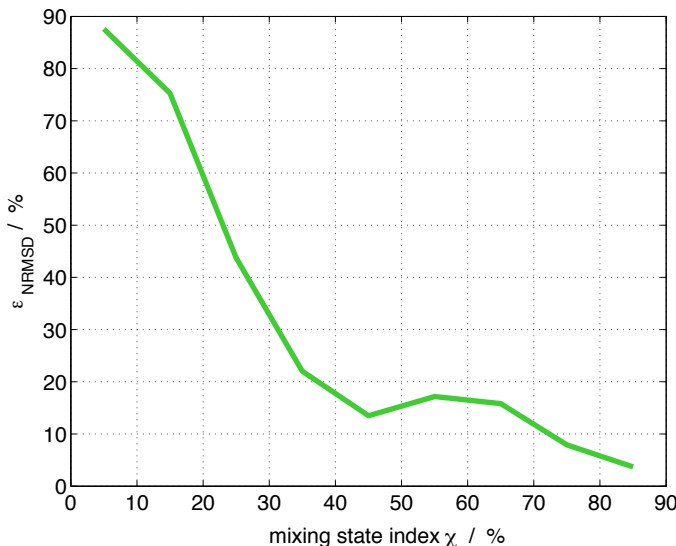

**Figure 9.** Normalized root mean square deviation in CCN concentration due to composition averaging ($\epsilon_{\mathrm{NRMSD}}$) as a function of mixing state index $\chi$, see Equation 3 for definition.

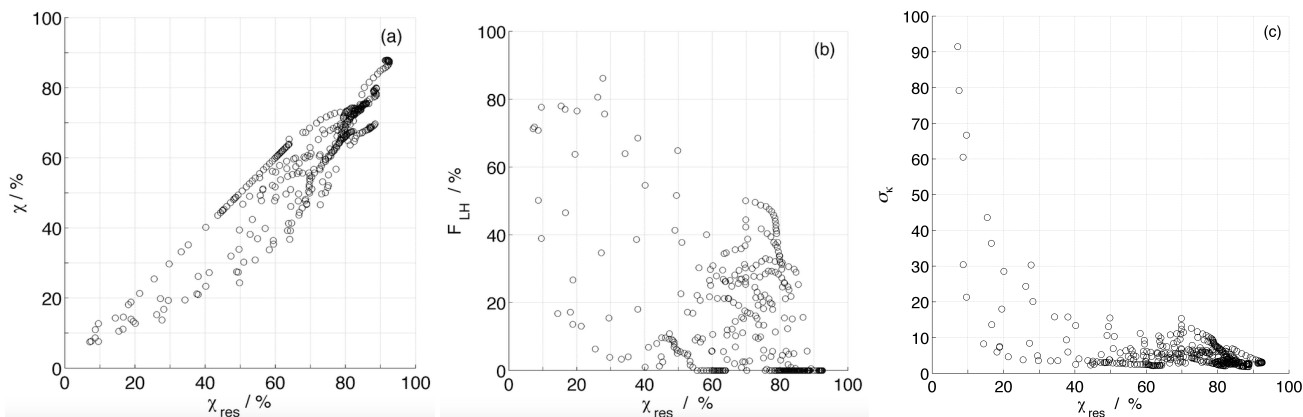

**Figure 10.** Relationship of (a) size-restricted mixing state parameter $\chi_{\mathrm{res}}$ and mixing state parameter, $\chi$, (b) size-restricted mixing state parameter $\chi_{\mathrm{res}}$ and number fraction of particles with low hygroscopicity, $F_{\mathrm{LH}}$, and (c) size-restricted mixing state parameter $\chi_{\mathrm{res}}$ and geometric standard deviation of the $\kappa$-distribution, $\sigma_{\kappa}$, for all 384 aerosol populations in $\mathbb{P}$.

## 6 Relationship of $\chi$ to other metrics of hygroscopic mixing state

The mixing state index $\chi$ can be interpreted as a measure of the heterogeity of a particle distribution with respect to its hygroscopic properties. In Section 5 we showed how $\chi$ relates to the error in CCN concentration that is introduced by assuming that the particle population is internally mixed. To illustrate the concept further, in this section we will show how $\chi$ relates to

other possible metrics of hygroscopicity heterogenity, and how each of them relates to the error in CCN due to neglecting mixing state information.

We focus here on three other metrics: (1) A size-restricted version of $\chi$, $\chi_{\mathrm{res}}$, only considers particles in the diameter size range of 30 nm to 150 nm when calculating the mixing state index. The rationale for this is that this constitutes the size range for which composition matters most because particles smaller than 30 nm are unlikely to activate, and particles larger than 150 nm

are very likely to activate, regardless of their composition. (2) The number fraction of particles with low hygroscopicity, $F_{\mathrm{LH}}$. Here we considered particles with $\kappa < 0.1$ as having low hygroscopicity. (3) The geometric standard deviation of the aerosol distribution with respect to the hygroscopicity parameter $\kappa$, $\sigma_{\kappa}$. The choice of this parameter was motivated by Su et al. (2010) who used the concept of hygroscopicity distribution to characterize aerosol particle mixing state with regard to CCN properties. For consistency, $\sigma_{\kappa}$ and $F_{\mathrm{LH}}$ were also evaluated using only particles with diameters between 30 and 150 nm.

Figure 10a shows that the mixing state index $\chi$ for the entire population and the size-restricted mixing state index $\chi_{\mathrm{res}}$ are strongly correlated. The sub-population with particle diameters between 30 and 150 nm is in many cases more internally mixed than the whole population. From Figure 10b we conclude that, at least for our study, populations with a high fraction of low-hygroscopic particles tend to be associated with low values of $\chi_{\mathrm{res}}$, and vice versa. However, the scatter is large, and populations can be found that do not contain particles in the low-hygroscopicity category, but exhibit a wide range of mixing

state index values. It is also plausible that populations exist that are internally mixed, i.e. have high mixing state index, but

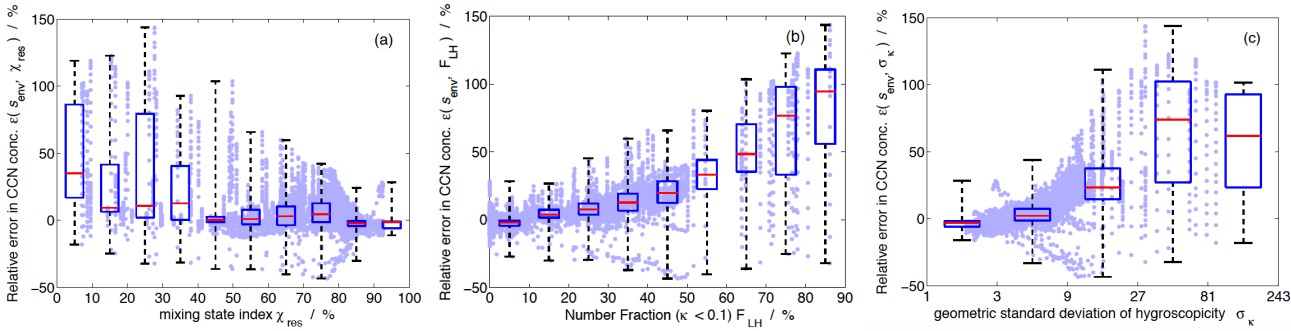

**Figure 11.** Relative error of (a) $\epsilon(s_{\mathrm{env}}, \chi_{\mathrm{res}})$, (b) $\epsilon(s_{\mathrm{env}}, F_{\mathrm{LH}})$, and (c) $\epsilon(s_{\mathrm{env}}, \sigma_\kappa)$ for each individual aerosol population in $\mathbb{P}$ (light purple dots), evaluated for 20 supersaturation values between 0.05% and 1%, analogous to Figure 6.

contain mainly non-hygroscopic species, i.e. have a high value of $F_{\mathrm{LH}}$. The fact that we do not show such populations in our study is due to our set up, where a hygroscopic background population is always present into which we emit fresh, less-hygroscopic particles. Lastly, Figure 10c shows that more externally mixed populations (small $\chi_{\mathrm{res}}$) are associated with large values of $\sigma_\kappa$.

5   Figure 11 relates the relative error in CCN concentration to the mixing state metrics $\chi_{\mathrm{res}}$, $F_{\mathrm{LH}}$, and $\sigma_\kappa$, analogous to Figure 6. These figures looks qualitatively similar, which is expected given that all these measures are measures of heterogeneity of the population with respect to hygroscopic properties. However, fundamentally, the mixing state index $\chi$ is appealing because it has a defined range (from 0 to 100%), and the property that when $\chi$ is 100%, the population is perfectly internally mixed with a CCN error of zero. It can also be easily generalized to alternate mixing state definition, e.g. based on optical properties, by

10   defining the surrogate species appropriately.

The measure $F_{\mathrm{LH}}$ also has a defined range (from 0 to 100%), and a value of 0 implies that the population is fairly (although not necessarily totally) homogeneous in the sense that all particles have a hygroscopicity larger than 0.1. However, the reverse implication does not hold. If the population is homogeneous (perfectly internally mixed), $F_{\mathrm{LH}}$ could be 0 or 100%. The geometric standard deviation $\sigma_\kappa$ is 1 for all perfectly internally mixed populations, but does not have a defined upper limit.

## 15  7   Conclusions

Our analysis used particle-resolved modeling and a metric for aerosol mixing state to develop a framework for quantifying error in CCN activity due to simplifying assumptions about mixing state. Mixing state information does indeed become unimportant for more internally-mixed populations. The NRMSD error decreased to less than 10% when $\chi$ was larger than 75%. For more externally-mixed populations ($\chi$ below 60%), a wide range of error in CCN concentration existed, with errors as large as 150% for $\chi$ lower than 20%. Because the composition averaging was performed for the whole population, without retaining any size-resolved composition information, the maximum CCN error as a function of $\chi$ gives an upper bound of CCN error. It should

be noted that even for low $\chi$ values, results close to the true result may be obtained because of error cancellations. In summary, the CCN error depends on both $\chi$ and $s_{\mathrm{env}}$. As the population becomes more internally-mixed ($\chi$ approaching 100%), the CCN error is small regardless of the $s_{\mathrm{env}}$ threshold. For the aerosol populations in this study, we found higher sensitivity to $\chi$ at high $s_{\mathrm{env}}$ threshold and lower sensitivity to $\chi$ at low $s_{\mathrm{env}}$ threshold.

We also explored the relationship of CCN error to other measures of mixing state, specifically a size-restricted $\chi$, the fraction of particles with low hygroscopicity, and the geometric standard deviation of the $\kappa$-distribution. These other measures also capture aspects of the heterogeneity of the particle population and the dependence of CCN error on these quantities are qualitatively similar to the one when using $\chi$. However, $\chi$ has advantages as a mixing state metric due to its defined range (0 to 100%) and well-defined extremes (0% is fully internally mixed and 100% is full externally mixed).

Several avenues exist for applying and extending our approach in future work. To confirm the validity of this approach, it would be useful to perform CCN closure studies on field campaign data with different mixing state assumptions, and to obtain concurrent aerosol diversity measurements, using for example single-particle mass spectrometry or off-line single-particle analytical techniques. The same framework, with appropriately defined surrogate species, could also be applied to quantify how error for aerosol optical properties depends on aerosol mixing state.

Aerosol-cloud interactions continue to be one of the major sources of future climate prediction uncertainties (IPCC, 2013). One of the reasons for this is the challenge to represent the many microscale aerosol processes in large-scale global climate models (Seinfeld et al., 2016). Using a high-fidelity aerosol model, our study provides quantitative support that mixing state is important for determining the aerosol impact on clouds. As a consequence we conclude that the aerosol representation in global models should account for mixing state. The aerosol microphysics schemes in several current global climate models

have been moving into this direction (Ghan and Schwartz, 2007; Bauer et al., 2008; Jacobson, 2002; Aquila et al., 2011; Kaiser et al., 2014; Mann et al., 2010; Liu et al., 2016; Stier et al., 2005) by separating the particles into interacting subpopulations, e.g. freshly emitted BC and aged BC. The central question is whether this degree of resolution of mixing state is sufficient to accurately predict CCN concentration and aerosol optical properties, and whether this framework allows for an accurate prediction of the mixing state *evolution*. In situ measurements of size-resolved aerosol mixing state, as suggested by Seinfeld

et al. (2016), are needed to constrain our models and the estimates of aerosol-cloud forcing in climate models.

*Author contributions.* J. Ching and N. Riemer designed the particle-resolved model simulations and analyzed the model data. M. West developed the PartMC model code. M. West and J. Fast contributed to the study design and the interpretation of the results. N. Riemer and J. Ching prepared the manuscript with contributions from all co-authors.

*Acknowledgements.* The authors thank Dr. Alla Zelenyuk for constructive comments and Dr. Kai Zhang for internal reviewing. Nicole
Riemer acknowledges NSF AGS CAREER grant 1254428 and EPA grant 83504201. The contents are solely the responsibility of the grantee and do not necessarily represent the official views of the US EPA. Further, US EPA does not endorse the purchase of any commercial products or services mentioned in the publication. Matthew West acknowledges NSF CMMI CAREER grant 1150490, and DOE ASR grant DE-SC0011771. Joseph Ching and Jerome Fast were supported by the U.S. Department of Energy (DOE) under the auspices of the

Atmospheric System Research (ASR) program of the Office of Science. Pacific Northwest National Laboratory is operated for DOE by Battelle Memorial Institute under contract DE-AC05-76RL01830.

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
