# Peer review of "Metrics to quantify the importance of mixing state for CCN activity"

_Atmospheric Chemistry and Physics, 2016_

## Referee Comment (RC1) · Anonymous Referee #1 · 16 Feb 2017

The manuscript with the title "Metrics to quantify the importance of mixing state for CCN activity" presents a study of simulated CCN activity with and without the consideration of the aerosol mixing state. Resulting deviations are studied as a function of mixing state and environmental supersaturation. It also presents a boundary to errors that need to be considered when neglecting mixing state information. The article is well written, conclusions are valid, and the content is scientifically relevant in the scope of ACP. I therefore recommend publication after the following comments have been addressed:

1. Figure 1 and Table 1 and 2 are 100% identical to figures and tables in Riemer et. al. 2013. Even with the authors and the Journal being the same I would consider this a questionable practice. The least the authors can do is to add a note to the captions saying "Taken from Riemer et. al. 2013".

2. Most presented findings are statistical observations. I am convinced it would add great value to the manuscript if the authors discussed potential physical origins of their findings. E.g. why do different populations experience different amounts of error cancellations (page 12, line ∼9)? What physical characteristic might explain the dependence of the relative error on supersaturation threshold (page 12&13)?

3. Figure 7: the x-axis looks like a ratio rather then %. Also, the caption mentions insets which do not seam to be there.

---

## Referee Comment (RC2) · Anonymous Referee #2 · 7 Mar 2017

**Comment on "Metrics to quantify the importance of mixing state for CCN activity" by Ching et al.**

This manuscript presented particle-resolved model simulations to quantify error in CCN predictions when mixing state information is neglected. The authors classified the chemical model species according to hygroscopicity, defining two surrogate species, a low hygroscopicity group (black carbon and primary organic aerosol) and a high hygroscopicity group (inorganic and secondary organic aerosol species), and investigated the error for eight urban plume scenarios. The error was analyzed against the mixing state index ($\chi$), a parameter proposed by Riemer and West (ACP, 13, 11423-11439, 2013) to describe the degree of external and internal mixing of aerosol populations. The results show that neglecting the mixing state information has less influence on the CCN prediction for more internally-mixed aerosol populations than more externally mixed aerosol particles. The relationship of $\chi$ and the error in CCN predictions is not unique and the reasons have been discussed. I would recommend publication if the authors could address my comments as listed below.

**General comments**

1. Importance of the mixing state metrics

My major concern is how the new metrics ($\chi$) will help to quantify the mixing state effect? In this study, the determination of mixing state effect was done by comparing CCN predictions of cases with and without composition averaging. If I understand correct, it means that the mixing state effect is determined without the metric $\chi$. So, why would we need such a parameter if it is not even used?

2. Performance of the mixing state metrics

One of my questions during my reading is that if a single $\chi$ corresponds to a unique error in CCN predictions and if it can be used in the CCN prediction or even better than existing parameters. The authors answered my first question, and showed that the relationship of $\chi$ and the error in CCN predictions is not unique. According to the size-resolved hygroscopicity distribution in Fig. 4, there are two kappa modes and my feeling is that the fraction of the low hygroscopic mode ($F_{LH}$) is a critical parameter for the errors when neglecting the mixing state information. Could you make similar plots as in Fig. 6 and Fig.7 but using $F_{LH}$ instead of $\chi$? If the error shows more converged dependence on $F_{LH}$, $\chi$ may not be a better parameter for the CCN prediction. Besides, $\chi$ is hard to determine in practice by available measurement techniques.

3. Comparison of $\chi$ to existing parameters

$\chi$ is a single parameter containing more intensive information. The authors have nicely presented its general concept by a nice illustration of Fig. 1. But it is still hard to fully understand it. Can you plot the series of $\chi$ and compared it to other well-established parameters, e.g., $F_{LH}$, or the (geometric) standard deviation of kappa distribution, etc.? Does a higher $\chi$ correspond to a larger $F_{LH}$ or a smaller a standard deviation? The potential link to other mixing state parameter may help people to accept the new parameter.

4. Design of experiments and discussions

In this study, the performance of $\chi$ is evaluated by comparing the error with kinds of averaged diversity value over the whole size range. I suggest the authors to reconsider this. The errors in CCN prediction are controlled by multiple parameters, i.e., the evaluated supersaturation, the size distribution and the kappa distribution. We know that the particle size has a dominant effect on the CCN activation. But if we want to quantify the effects of particle size on the CCN prediction, can we plot the error against the averaged particle size as what was done for $\chi$?

It is not clear what's the better solution but maybe if the authors could try to used size-resolved $\chi$ and check how to use it in CCN prediction or parameterization, e.g., maybe there is a compact empirical relation between $\chi$ and the averaged activation fraction at each size.

**Minor comments:**

**Abstract** "However, it has been difficult to rigorously investigate this assumption because appropriate metrics for mixing state were lacking"

I think the kappa distribution and the corresponding parameters (mean kappa, mode kappa, and standard deviation) in Su et al. (2010) may be as good as $\chi$ in representing the CCN-relevant mixing state.

**Page 8 ln 10,**

Can the authors specify which kappa values were used for the two surrogate groups and how to calculate kappa for internally mixed particle?

**Ref:**

Su, H., Rose, D., Cheng, Y., Gunthe, S., Massling, A., Stock, M., Wiedensohler, A., Andreae, M., and Poschl, U.: Hygroscopicity distribution concept for measurement data analysis and modeling of aerosol particle mixing state with regard to hygroscopic growth and CCN activation, Atmospheric Chemistry and Physics, 10, 7489-7503, 10.5194/acp-10-7489-2010, 2010.

---

## Author Comment (AC1) · 2 May 2017

Joseph Ching
ppching@gmail.com

May 2, 2017

Holger Tost
Editor, Atmospheric Chemistry and Physics

**Re: Response to Reviews**
Manuscript Number: acp-2016-1044
Manuscript Title: Metrics to quantify the importnce of mixing state for CCN activity
Manuscript Authors: J. Ching, J. Fast, M. West, and N. Riemer

**1 Response to Reviewer #1's comments**

We greatly appreciate the reviewer's comments. We revised the manuscript accordingly with changes marked in blue. Our responses are as follows:

**(1.1)** Figure 1 and Table 1 and 2 are 100% identical to figures and tables in Riemer et. al. 2013. Even with the authors and the Journal being the same I would consider this a questionable practice. The least the authors can do is to add a note to the captions saying "Taken from Riemer et. al. 2013".

> We updated the captions to Figure 1 and Tables 1 and 2 accordingly.
>
> We also added text to emphasize that we summarize the important key points from Riemer and West (2013) for the convenience of the reader:
>
> - page 3, line 4: "The salient points are summarized as follows."
> - page 3, lines 7–8: "From this quantity, all other mass-related quantities can be defined, as detailed in Riemer and West (2013) and here listed in Table 1, and the diversity metrics can be constructed as shown in Table 2."

**(1.2)** Most presented findings are statistical observations. I am convinced it would add great value to the manuscript if the authors discussed potential physical origins of their findings. E.g. why do different populations experience different amounts of error cancellations (page 12, line 9)? What physical characteristic might explain the dependence of the relative error on supersaturation threshold (page 12&13)?

> We have made a number of changes and clarifications to the paper in an attempt to explain the physical origins of our findings. We renamed Section 5 to "Relationship of error in CCN concentration and mixing state index $\chi$" to highlight that this is where we will discuss this relationship. In this section we discuss Figure 5, which clarifies that "Error is caused only by the *difference* in the number of particles that activate in the reference case, but do not activate after composition averaging [...], and the cases that activate after composition averaging, but did not activate in the reference case."
>
> In this section we also explain the physical basis of zero error for internally mixed populations as the fact that "For populations with $\chi = 100\%$, the particles' composition is the same across the entire population, and hence [...] the number concentrations in quadrants B and D are zero." Correspondingly, we explain the physical basis of the error for partially-internally mixed populations as depending on "both the extent of distortion of the critical supersaturation distribution and the supersaturation threshold."
>
> To explain the dependence of the relative error on the supersaturation threshold, we clarified that "for a given supersaturation threshold and a given $\chi$ value, different populations experience different amounts of the error cancellations" described above. A detailed investigation of this is provided by Figure 8, which together with Figure 5 provides physical understanding of the cancellation phenomenon. We explained in the paper that "the change in particle number concentration per supersaturation interval is smaller for $s_c$ of 0.1% compared to $0.3 < s_c < 1.5\%$" due to the shape of the population $\mathbb{P}$ in Figure 5.

**(1.3)** Figure 7: the x-axis looks like a ratio rather then %. Also, the caption mentions insets which do not seem to be there.

> Thanks for pointing this out. We fixed both of these issues.

**References**

N. Riemer and M. West. Quantifying aerosol mixing state with entropy and diversity measures. *Atmos. Chem. Phys.*, 13:11423–11439, 2013.

---

## Author Comment (AC2) · 2 May 2017

Joseph Ching
ppching@gmail.com

May 2, 2017

Holger Tost
Editor, Atmospheric Chemistry and Physics

**Re: Response to Reviews**
Manuscript Number: acp-2016-1044
Manuscript Title: Metrics to quantify the importnce of mixing state for CCN activity
Manuscript Authors: J. Ching, J. Fast, M. West, and N. Riemer

**1 Response to Reviewer #2's comments**

We thank the reviewer for their comments and suggestions. We revised the manuscript accordingly with changes marked in blue. Our responses are as follows:

**(2.1)** Importance of the mixing state metrics: My major concern is how the new metrics ($\chi$) will help to quantify the mixing state effect? In this study, the determination of mixing state effect was done by comparing CCN predictions of cases with and without composition averaging. If I understand correct, it means that the mixing state effect is determined without the metric $\chi$. So, why would we need such a parameter if it is not even used?

> The reviewer is correct—since we have all the per-particle information we can determine the error, and if this was all we wanted to do then the metric $\chi$ (or any other mixing state metric) is not needed.
>
> However, our goal is to relate the error in CCN concentration due to the internal mixture assumption to a quantitative measure of mixing state. The paragraph in the introduction, p. 2, line 13 states this goal, and we added text to clarify this further, p. 2, lines 18–20: "The central question that we address is: For aerosol populations of a given mixing state, what magnitude of errors can we expect for estimating CCN concentrations when assuming that the population is internally mixed?"

**(2.2)** Performance of the mixing state metrics: One of my questions during my reading is that if a single $\chi$ corresponds to a unique error in CCN predictions and if it can be used in the CCN prediction or even better than existing parameters. The authors answered my first question, and showed that the relationship of $\chi$ and the error in CCN predictions is not unique. According to the size-resolved hygroscopicity distribution in Fig. 4, there are two kappa modes and my feeling is that the fraction of the low hygroscopic mode ($F_{LH}$) is a critical parameter for the errors when neglecting the mixing state information. Could you make similar plots as in Fig. 6 and Fig. 7 but using $F_{LH}$ instead of $\chi$? If the error shows more converged dependence on $F_{LH}$, $\chi$ may not be a better parameter for the CCN prediction. Besides, $\chi$ is hard to determine in practice by available measurement techniques.

> Thanks for this suggestion, which together with reviewer's point 2.3 inspired us to add another section to the paper. We think that including $F_{LH}$ and the geometric standard deviation of the $\kappa$-distribution in the discussion will answer the questions that many readers might have.
>
> The new section (Section 6) is titled "Relationship of $\chi$ and CCN error to other metrics of hygroscopic mixing state", and we added two figures.
>
> Figure 10 shows the relationship of (a) size-restricted mixing state parameter $\chi_{res}$ and mixing state parameter, $\chi$, (b) size-restricted mixing state parameter $\chi_{res}$ and number fraction of particles with low hygroscopicity, $F_{LH}$, and (c) size-restricted mixing state parameter $\chi_{res}$ and geometric standard deviation of the $\kappa$-distribution, $\sigma_\kappa$, for all 384 aerosol populations in $\mathbb{P}$.
>
> Figure 11 is analogous to Figure 6 and shows (a) Relative error $\epsilon(s_{env}, \chi_{res})$, (b) relative error $\epsilon(s_{env}, F_{LH})$, and (c) Relative error $\epsilon(s_{env}, \sigma_\kappa)$ for each individual aerosol population in $\mathbb{P}$, evaluated for 20 supersaturation values between 0.05% and 1%.
>
> We also added text in the conclusions, p. 18, lines 19–23: "We also explored the relationship of CCN error other measures of mixing state, specifically a size-restricted $\chi$, the fraction of particles with low hygroscopicity, and the geometric standard deviation of the $\kappa$-distribution. These other measures also capture aspects of the heterogeneity of the particle population and the dependence of CCN error on these quantities are qualitatively similar to the one when using $\chi$. However, $\chi$ has advantages as a mixing state metric due to its defined range (0 to 100%) and well-defined extremes (0% is fully internally mixed and 100% is full externally mixed)."

**(2.3)** Comparison of $\chi$ to existing parameters: $\chi$ is a single parameter containing more intensive information. The authors have nicely presented its general concept by a nice illustration of Fig. 1. But it is still hard to fully understand it. Can you plot the series of $\chi$ and compared it to other well-established parameters, e.g., $F_{\mathrm{LH}}$, or the (geometric) standard deviation of kappa distribution, etc.? Does a higher $\chi$ correspond to a larger $F_{\mathrm{LH}}$ or a smaller standard deviation? The potential link to other mixing state parameter may help people to accept the new parameter.

> This was addressed in conjunction with the response to comment (2.2), and forms the content of the new Section 6.

**(2.4)** Design of experiments and discussion: In this study, the performance of $\chi$ is evaluated by comparing the error with kinds of averaged diversity value over the whole size range. I suggest the authors to reconsider this. The errors in CCN prediction are controlled by multiple parameters, i.e., the evaluated supersaturation, the size distribution and the kappa distribution. We know that the particle size has a dominant effect on the CCN activation. But if we want to quantify the effects of particle size on the CCN prediction, can we plot the error against the averaged particle size as what was done for $\chi$? It is not clear what's the better solution but maybe if the authors could try to used size-resolved $\chi$ and check how to use it in CCN prediction or parameterization, e.g., maybe there is a compact empirical relation between $\chi$ and the averaged activation fraction at each size.

> We did explore what the error distribution would look like if size-resolved composition information was retained. This is described on p. 14, lines 17–26. We did not include a figure for these results because they are qualitatively similar to Figure 6.
>
> In the new Section 6, we now also added some material to show what happens if $\chi$ is calculated based on a size restricted population (Figure 10a and 11a, see response to comment (2.2) for details). Again, the error distribution looks qualitatively very similar to Figure 6.

**(2.5)** Abstract "However, it has been difficult to rigorously investigate this assumption because appropriate metrics for mixing state were lacking"

I think the kappa distribution and the corresponding parameters (mean kappa, mode kappa, and standard deviation) in Su et al. (2010) may be as good as $\chi$ in representing the CCN-relevant mixing state.

> We agree with the reviewer and removed this sentence.

**(2.6)** Page 8 ln 10, Can the authors specify which kappa values were used for the two surrogate groups and how to calculate kappa for internally mixed particle?

> We added the specifics about this on p. 8, lines 11/12– p. 9, lines 1–4:
>
> "Since we track the composition evolution of each individual particle throughout the simulation, we can calculate the critical supersaturation $s_{\mathrm{c}}$ for each particle as described in Riemer et al. (2010), using the concept of the dimensionless hygroscopicity parameter $\kappa$ (Petters and Kreidenweis, 2007). The overall $\kappa$ for a particle is the volume-weighted average of the $\kappa$ values of the constituent species. Based on Petters and Kreidenweis (2007) we assume $\kappa = 0.65$ for all salts formed from the $\mathrm{NH_4^+ - SO_4^{2-} - NO_3^-}$ system. For all MOSAIC model species that represent SOA we assume $\kappa = 0.1$, and for POA and BC we assume $\kappa = 0.001$ and $\kappa = 0$, respectively."
>
> Note that we do not assign kappa values for the surrogate species as such, but calculate the overall $\kappa$ for a particle as the volume-weighted average of the $\kappa$ values of the constituent species, and assign $\kappa$ values of the constituent species as specified above.

**References**

M. D. Petters and S. M. Kreidenweis. A single parameter representation of hygroscopic growth and cloud condensation nucleus activity. *Atmos. Chem. Phys.*, 7:1961–1971, 2007.

N. Riemer, M. West, R. Zaveri, and R. Easter. Estimating black carbon aging time-scales with a particle-resolved aerosol model. *J. Aerosol Sci.*, 41:143, 2010.